# Rabbit Meat Extract Induces Browning in 3T3−L1 Adipocytes via the AMP−Activated Protein Kinase Pathway

**DOI:** 10.3390/foods12193671

**Published:** 2023-10-06

**Authors:** In-Seon Bae, Jeong Ah Lee, Soo-Hyun Cho, Hyoun-Wook Kim, Yunseok Kim, Kangmin Seo, Hyun-Woo Cho, Min Young Lee, Ju Lan Chun, Ki Hyun Kim

**Affiliations:** 1Animal Products Utilization Division, National Institute of Animal Science, Rural Development Administration, Wanju 55365, Republic of Korea; j2970703@rda.go.kr (J.A.L.); shc0915@rda.go.kr (S.-H.C.); woogi78@rda.go.kr (H.-W.K.); kys88@rda.go.kr (Y.K.); 2Department of Animal Resources Science, Kongju National University, Yesan 32439, Republic of Korea; 3Animal Welfare Research Team, National Institute of Animal Science, Rural Development Administration, Wanju 55365, Republic of Korea; seokm@rda.go.kr (K.S.); jhwoo3856@rda.go.kr (H.-W.C.); mylee1231@rda.go.kr (M.Y.L.); julanchun@rda.go.kr (J.L.C.); kihyun@rda.go.kr (K.H.K.)

**Keywords:** rabbit meat, AMPK pathway, adipocytes, browning, obesity

## Abstract

The browning of white adipocytes may be an innovative approach to address obesity. This study investigated the effects of rabbit meat extract on 3T3−L1 adipocytes, with a specific emphasis on inducing browning. The browning effects of rabbit meat extract were evaluated by analyzing genes specifically expressed in 3T3−L1 adipocytes using quantitative PCR and immunoblotting. Rabbit meat extract increased the expression of brown adipocyte−specific markers, UCP1 and PGC1α, and mitochondrial biogenesis factors, TFAM and NRF1, without affecting cell viability in fully differentiated 3T3−L1 adipocytes. Moreover, adipocyte differentiation and the triglyceride content were decreased; hormone−sensitive lipase activity was promoted. Rabbit meat extract activated the AMPK pathway in the differentiated 3T3−L1 cells. However, in adipocytes treated with rabbit meat extract, the expression of genes related to browning was reduced by the AMP−activated protein kinase (AMPK) inhibitor, dorsomorphin dihydrochloride. To the best of our knowledge, this is the first study to demonstrate that rabbit meat extract induces the browning of white adipocytes via the activation of the AMPK pathway, thereby demonstrating its therapeutic potential in preventing obesity.

## 1. Introduction

Obesity, caused by an imbalance in energy storage and expenditure, is closely related to the development of chronic diseases, such as type 2 diabetes and cardiovascular disease [1,2]. Therefore, controlling or preventing obesity is necessary to avoid these diseases. Increases in the adipocyte number and size are major causes of obesity [3,4]. The mammalian adipose tissue is divided into white and brown adipocytes [5,6]. White adipocytes store energy in the form of triglycerides, which are stored in unilocular lipid droplets, whereas brown adipocytes have multilocular lipid droplets and abundant mitochondria that dissipate energy [7]. A third type, termed as beige adipocytes, has the characteristics of brown adipocytes and is formed in white adipose tissue [7,8]. The ability of white adipocytes to convert to the brown adipocyte phenotype in response to specific stimuli, such as exposure to cold temperatures or beta−adrenergic stimulation, is being studied for obesity treatment [9,10]. Beige adipocytes have the brown adipocyte phenotype with a high expression of genes related to mitochondrial biogenesis, brown−fat−specific markers uncoupling protein 1 (UCP1), and peroxisome proliferator−activated receptor γ coactivator 1 α (PGC−1α) [7,8,9,10]. Currently, research on detecting substances and regulatory factors to induce beige adipocytes is being performed.

AMP−activated protein kinase (AMPK) plays a central role in the conversion of white adipocytes into brown adipocytes [11,12]. Certain foods have the ability to promote the browning of white adipocytes by activating AMPK. The combination of *Panax ginseng* and *Diospyros kaki* leaf has the potential to inhibit adipogenesis and induce the browning process in adipocytes through the activation of AMPK [13]. Black wheat extract and strawberry methanolic extract have regulatory effects of browning in 3T3−L1 adipocytes through the activation of the AMPK pathway [14,15]. The activation of AMPK leads to increased thermogenesis, mitochondrial biogenesis, and fatty acid oxidation in adipocytes. In addition, AMPK also regulates inflammatory cascades to improve chronic inflammatory diseases [16,17]. Thus, therapeutics aimed at activating the AMPK pathway is a promising approach in obesity and metabolic dysfunction.

Several studies using meat−derived extracts are being conducted. Goat meat extract induces apoptosis in AGS and HT−29 cells by increasing the expression of tumor−suppressor genes, p21 and p53, and shows anti−muscular atrophy activity in dexamethasone−treated C2C12 myocytes [18]. Chicken meat extract has beneficial effects on cognitive ability and memory in middle−aged mouse, while also reducing inflammation and oxidative stress in Raw264.7 cells [19,20,21]. Pork extract strengthens the immune system of mice, and beef extract induces myotube hypertrophy and myoblast proliferation in C2C12 cells [22,23]. However, there is a lack of adequate research regarding the physiological effects of meat−derived extracts in comparison to natural products or extracts derived from plants. In particular, the physiological efficacy of rabbit meat has not been reported. Therefore, this study aimed to investigate the anti−adipogenic effect of rabbit meat extract on 3T3−L1 adipocytes and to understand the underlying mechanisms by focusing on AMPK activity. 

## 2. Materials and Methods

### 2.1. Rabbit Meat Preparation

The rabbit meat was purchased from a local market in Sangju, Republic of Korea. Distilled water (2 L) was added to rabbit meat (20 kg) in a non−woven bag and heated at 120 °C for 12 h. When the target temperature reached 120 °C, the pressure was maintained at 30 psi. After 12 h, the pressure was removed and the non−woven bag containing the rabbit meat was compressed. The rabbit meat extract was centrifuged at 3000× *g* at 25 °C for 10 min, and the supernatant was filtered through a No. 1 filter paper. The rabbit meat extract was freeze−dried and used in the experiment. 

### 2.2. Cell Culture and Differentiation

3T3−L1 preadipocytes were cultured in Dulbecco’s modified Eagle’s medium (DMEM; Welgene, Daegu, Republic of Korea) supplemented with 10% bovine calf serum and 1% penicillin−streptomycin (Hyclone, Logan, UT, USA) at 37 °C in a humidified atmosphere of 5% carbon dioxide until confluence. To induce differentiation, two days after 100% confluence, the medium was replaced with a differentiation medium containing 10% fetal bovine serum (FBS; Hyclone), 0.5 mM isobutylmethylxanthine (Sigma−Aldrich, St. Louis, MO, USA), 1 μM dexamethasone (Sigma−Aldrich), and 10 μg/mL insulin (Sigma−Aldrich). After two days, the cells were cultured in DMEM supplemented with 10% FBS and 10 μg/mL insulin for an additional two days. Subsequently, the medium was replaced with DMEM containing 10% FBS every 48 h until day 8. Confluent pre−adipocytes were treated with rabbit meat extract and differentiation medium to induce differentiation for eight days.

### 2.3. Cell Viability Assay

Cell viability was measured using EZ−CyTox (Daeil Lab Service, Seoul, Republic of Korea). 3T3−L1 preadipocytes were seeded into a 96−well plate at 2 × 10^4^ cells per well. After treating the cells with rabbit meat extract (10 μg/mL to 200 μg/mL), the DMEM medium in each well was replaced with 0.01 mL EZ−Cytox solution and then incubated at 37 °C for 1 h. Absorbance was measured at 450 nm using a microplate reader.

### 2.4. Oil Red O Staining

Intracellular lipid accumulation was measured using oil red O staining. 3T3−L1 adipocytes were fixed with 4% paraformaldehyde at room temperature for 20 min and gently washed three times with phosphate−buffered saline (PBS; Welgene). The cells were stained with 0.6% oil red O dye for 1 h at room temperature, and then washed three times with distilled water. Images were captured using a microscope, and 100% isopropanol was added for 15 min to quantitatively analyze staining results. The dissolved oil red O content was measured via absorbance at 500 nm using a microplate reader. 

### 2.5. RNA Extraction and Quantitative Reverse−Transcription Polymerase Chain Reaction (qRT−PCR)

Total RNA was extracted from 3T3−L1 adipocytes using Trizol reagent (Sigma−Aldrich). Then, cDNA was synthesized using iScript cDNA synthesis kit (Bio−Rad, Hercules, CA, USA) and 1 μg of RNA was used for reverse transcription as per manufacturer’s instructions. The mRNA expression level was measured via quantitative real−time PCR using SsoAdvanced universal SYBR Green supermix (Bio−Rad) and a 7500 real−time PCR system (Applied Biosystems, Foster City, CA, USA) as per manufacturer’s instructions. The qRT−PCR was conducted under the following conditions: 95 °C for 30 s, followed by 40 cycles at 95 °C for 5 s and 60 °C for 30 s, and then at 95 °C for 15 s. 

The PCR primer sequences were as follows: forward, 5′−GTG ACG TTG ACA TCC GTA AAG A−3′ and reverse, 5′−GCC GGA CTC ATC GTA CTC C−3′ for β−actin; forward, 5′−TAT GGA GTG ACA TAG AGT GTG CT−3′ and reverse, 5′−CCA CTT CAA TCC ACC CAG AAA G−3′ for PGC1α; forward, 5′−TAT GGC GGA AGT AAT GAA AGA CG−3′ and reverse, 5′−CAA CGT AAG CTC TGC CTT GTT−3′ for nuclear respiratory factor 1 (NRF1); forward, 5′−ATT CCG AAG TGT TTT TCC AGC A−3′ and reverse, 5′−TCT GAA AGT TTT GCA TCT GGG T−3′ for mitochondrial transcription factor A (TFAM); forward, 5′−ACG TCC CCT GCC ATT ACT G−3′ and reverse, 5′−GGT ACG CTT GGG TAC TGT CC−3′ for UCP1; forward, 5′−TGT TGC CGG GGT CAT ATC CTA−3′ and reverse, 5′−AGC ATC GGG TAG TCG CCA TA−3′ for NADH:ubiquinone oxidoreductase subunit B8 (NDUFB8); forward, 5′−AAT TTG CCA TTT ACC GAT GGG A−3′ and reverse, 5′−AGC ATC CAA CAC CAT AGG TCC−3′ for succinate dehydrogenase iron−sulfur subunit, mitochondrial (SDHB); forward, 5′−AAA GTT GCC CCG AAG GTT AAA−3′ and reverse, 5′−GAG CAT AGT TTT CCA GAG AAG CA−3′ for ubiquinol−cytochrome c reductase core protein 2 (UQCRC2); forward, 5′−ATT GGC AAG AGA GCC ATT TCT AC−3′ and reverse, 5′−CAC GCC GAT CAG CGT AAG T−3′ for cytochrome c oxidase subunit IV (COXIV); forward, 5′−TCT CCA TGC CTC TAA CAC TCG−3′ and reverse, 5′−CCA GGT CAA CAG ACG TGT CAG−3′ for ATP synthase alpha subunit (ATP5A1); forward, 5′−GAG CAT GTT TTC AGA CGA CTT TG−3′ and reverse, 5′−CCG AGG GTC TTG ATG TTT CCT T−3′ for heat shock protein family B member 7 (HSPB7); forward, 5′−TCA CCC TCC CTT CAA ACT GTA−3′ and reverse, 5′−GTT TCA CTG CGG AGA TGA CAT−3′ for EBF transcription factor 3 (EBF3); forward, 5′−GAG GAC GAT TCC ACT GGA CTT GT−3′ and reverse, 5′−GTA ATG CTT TCC ACT GGA CTT GT−3′ for eosinophil−associated, ribonuclease A family, member 2 (EAR2); forward, 5′−AAC CTT GGA GTG AAG GAT CGC−3′ and reverse, 5′−GTA GGA GAG CCT ATT GGA GAT GT−3′ for CREB−binding protein/p300 interacting transactivator with Asp/Glu−rich C−terminal domain (CITED1); forward, 5′−ACG TCC CCT GCC ATT ACT G −3′ and reverse, 5′−GTA GGA GAG CCT ATT GGA GAT GT−3′ for TNF receptor superfamily member 9 (TNFRSF9); forward, 5′−TGT CAT CTG TGA AAA GGT GGT C−3′ and reverse, 5′−ACT GGA GCA GCG GTG TTA TG−3′ for TNF receptor superfamily member 5 (CD40). 

Relative mRNA expression level was analyzed using the delta delta Ct method with β−actin.

### 2.6. Western Blot Analysis

Cells were lysed using radioimmunoprecipitation assay buffer (RIPA buffer; 50 mM Tris−HCl, 150 mM sodium chloride (NaCl), 1% NP−40, 0.1% sodium dodecyl sulfate (SDS), a protease inhibitor cocktail, 50 mM sodium fluoride, and 0.2 M sodium orthovanadate). After centrifugation (12,000× *g* for 15 min), the protein concentration of the supernatant was measured. The proteins in the cell lysate were measured using the Braford assay. Next, 20 μg of protein from each sample was electrophoresed on 10% acrylamide SDS−polyacrylamide gel electrophoresis and transferred to a nitrocellulose membrane (Millipore, Burlington, MA, USA). Membranes were then blocked in 5% skimmed milk for 1 h at room temperature and incubated with primary and secondary antibodies. The primary antibodies were anti−UCP1 (Abcam, Waltham, MA, USA), anti−PGC1α (Boster Bio, Pleasanton, CA, USA), anti−adipose triglyceride lipase (ATGL; Boster Bio), anti−phospho hormone−sensitive lipase (HSL; Cell Signaling, MA, USA), anti−TFAM (Invitrogen, Carlsbad, CA, USA), anti−NRF1 (Invitrogen), anti−AMPK (Cell Signaling), anti−phospho−AMPK (Cell Signaling), anti−AKT (Cell Signaling), anti−phospho−AKT, and β−actin (Sigma−Aldrich); proteins were visualized using chemiluminescent HRP substrate (Advansta Inc., San Jose, CA, USA). 

### 2.7. Glucose Uptake

3T3−L1 pre−adipocytes were seeded in 6−well cell culture plates and adipocyte differentiation was induced in the presence or absence of rabbit meat extract (200 μg/mL) for eight days. The medium was replaced with serum−free DMEM and the cells were incubated for another 6 h. Cells were incubated with 1 μM insulin in Krebs−Ringer phosphate−HEPES (KRPH buffer; 1.2 mM potassium dihydrogen phosphate, 1.2 mM magnesium sulfate, 1.3 mM calcium chloride, 118 mM NaCl, 5 mM potassium chloride, and 30 mM Hepes; pH 7.5) containing 2% bovine serum albumin (Sigma−Aldrich) at 37 °C. Next, 2−deoxyglucose was added to the wells. After 20 min, the cells were washed with PBS containing 200 μM phloretin and lysed in ice−cold 10 mM Tris−HCl (pH 8.0). The uptake of 2−deoxyglucose was assessed using the 2−deoxyglucose uptake measurement kit (Cosmo Bio, Tokyo, Japan) as per manufacturer’s instructions. 

### 2.8. Statistical Analysis

All data were presented as mean ± SD of at least three independent experiments. Statistical analysis was performed using the SPSS program (SPSS Inc., Chicago, IL, USA). In the case of two experimental groups, the data were analyzed using Student’s *t*−test. In all data comparison analyses, *p*−values of less than 0.05 were considered statistically significant.

## 3. Results and Discussion

### 3.1. Rabbit Meat Extract Stimulated Browning in 3T3−L1 Adipocytes

This assessment of cytotoxicity in the rabbit meat extract represents their suitability for further applications. The cytotoxicity of the rabbit meat extract to 3T3−L1 pre−adipocytes and mature adipocytes was evaluated at concentrations from 10 to 200 μg/mL. As shown in Figure 1, treatment with various concentrations of rabbit meat extract in 3T3−L1 pre− and mature adipocytes did not differ from the control group in cell survival. Thus, the viability of 3T3−L1 cells is not affected by the rabbit meat extract. 

To investigate the induction of the browning effect of the rabbit meat extract, genes specifically expressed in differentiated adipocytes treated with rabbit meat extract were analyzed. The UCP1 and PGC1α are central players in the intricate metabolism of brown fat and its crucial roles in governing thermogenic processes, modulating energy dissipation, and contributing to overall metabolic homeostasis [24,25,26,27,28]. The mRNA levels of UCP1 and PGC1α were examined in the 3T3−L1 cells treated with rabbit meat extract. As shown in Figure 2A, the expression of UCP1 and PGC1α mRNA increased in a dose−dependent manner and significantly after the exposure to rabbit meat extract. The UCP1 and PGC1α protein levels were also upregulated in a rabbit−meat−extract−dependent manner (Figure 2B). In addition, the rabbit meat extract treatments significantly increased the expression levels of the markers of brown−like adipocytes (HSPB7, EBF3, EAR2, CITED1, and TNFRSF9; Figure 2C). These results suggest that rabbit meat extract can induce the conversion of white adipocytes into brown−like fat cells by upregulating the expression of thermogenic genes and white adipocyte browning−related genes without cytotoxicity.

For the first time, we confirmed that rabbit meat extract affects the expression of fat browning−related genes in adipocytes. In addition, the expression of these genes did not increase in the 3T3−L1 cells treated with goat meat, beef, and pork extracts. The rabbit meat extract may contain specific substances that induce the conversion of white adipocytes into brown adipocytes. We conducted a follow−up study aimed at investigating the specific substances within rabbit meat extract that induce the browning of adipocytes. 

### 3.2. Rabbit Meat Extract Suppressed Intracellular Lipid Accumulation in Adipocytes

The process of lipid accumulation is intricately linked to the onset and progression of obesity. We investigated the effect of rabbit meat extract on lipid accumulation in 3T3−L1 adipocytes. Through Oil Red O staining, a dose−dependent decrease in the lipid droplets was observed in the 3T3−L1 cells treated with rabbit meat extract (Figure 3A). As shown in Figure 3B, intracellular lipid accumulation was inhibited by approximately 50% in the 3T3−L1 adipocytes treated with 200 μg/mL of rabbit meat extract compared to the control group. The protein levels were measured using Western blotting to confirm the effect of the rabbit meat extract on lipolysis at the molecular level. ATGL is a lipase that acts on triacylglycerol and induces lipolysis [29]. HSL is the rate−limiting enzyme of lipid hydrolysis, and its phosphorylation plays an important role in lipolysis [29]. The expression levels of ATGL and HSL phosphorylated proteins in the 3T3−L1 adipocytes were also increased compared to the control group even after treatment with 10 μg/mL of rabbit meat extract (Figure 3C,D). Although the expression levels of these genes did not increase in a concentration−dependent manner, the lipolysis−related genes were upregulated (Figure 3C,D). These results indicate that the rabbit meat extract promotes lipolysis via increasing ATGL expression and inducing HSL phosphorylation. The application of rabbit meat extract in adipocytes presents evidence of its potential to inhibit the excessive formation of lipids within fat cells. 

### 3.3. Rabbit Meat Extract Promoted Mitochondrial Biogenesis in 3T3−L1 Adipocytes

As the browning of white adipocytes is closely related to increased mitochondrial activity, we investigated the stimulation of mitochondrial biogenesis via rabbit meat extract. First, we examined the expressions of TFAM and NRF−1, the representative genes of mitochondrial biogenesis. When the 3T3−L1 cells were treated with more than 100 μg/ml of rabbit meat extract, the expression levels of the TFAM and NRF1 genes were significantly increased (Figure 4A,B). To observe the changes in oxidative phosphorylation accompanying mitochondrial biogenesis, the subunit gene expression of each complex in the mitochondrial electron transport chain was investigated. The mRNA expression of NDUFB8 (Complex I), SDHB (Complex II), UQCRC2 (Complex III), COXIV (Complex IV), and ATP5A (Complex V) were increased as the rabbit meat extract concentration increased in the 3T3−L1 cells (Figure 4C). These results suggest that rabbit meat extract increases mitochondrial biogenesis in 3T3−L1 adipocytes. A higher mitochondrial biogenesis is associated with an increased energy expenditure. This means that cells with more mitochondria can burn more calories and generate more energy, contributing to the overall energy expenditure in the body [5,6,7,8]. This process is particularly characteristic of brown adipocytes, where mitochondria play important roles in energy metabolism and thermogenesis. 

### 3.4. Effect of Rabbit Meat Extract on Browning of Adipocytes Is Mediated via AMPK Pathway

The AMPK pathway is a major signaling factor in anti−obesity research. The activation of AMPK includes promoting the browning of adipocytes, inhibiting lipogenesis, and enhancing mitochondrial function. *Hovenia dulcis* branch extract releases lipid droplets via AMPK phosphorylation in 3T3−L1 cells and high−fat−diet−fed obese mice, and upregulates the expression of thermogenic genes such as UCP1 and PGC1 α [30]. In addition, Arriheuk extract reduces the triglyceride levels in 3T3−L1 adipocytes and the expression of proteins related to fat production; it induces mitochondrial biogenesis via the AMPK pathway [14]. Therefore, extracts derived from food sources have demonstrated anti−obesity effects by acting through the AMPK pathway. To determine whether the brown fat conversion effect of white fat in rabbit meat extract is induced by the AMPK pathway, we measured the protein levels of phosphorylated AMPK in 3T3−L1 cells after treatment with rabbit meat extract. Our results indicated that the phosphorylated AMPK protein levels increased as the rabbit meat extract concentration increased (Figure 5A). We then measured the expression of thermogenic genes and mitochondrial biogenesis factors via treatment with an AMPK inhibitor. The expression of thermogenic genes, UCP1 and PGC1α, and mitochondrial biogenesis factors, TFAM and NRF1, which were increased by the rabbit meat extract, decreased after treatment with the AMPK inhibitor compound C (Figure 5B). We confirmed that rabbit meat extract induces brown−like adipocytes through the AMPK pathway, as evidenced by the reduced expression of thermogenic genes and mitochondrial biogenesis factors when the AMPK pathway is inhibited, which is consistent with other studies on fat browning activated by AMPK. These results suggest that rabbit meat extract induces the browning of adipocytes via the AMPK pathway. 

### 3.5. Rabbit Meat Extract Improved Insulin Resistance in 3T3−L1 Adipocytes

The browning of white fat alleviates metabolic complications such as insulin resistance in type 2 diabetes mellitus. We investigated the effects of rabbit meat extract on glucose uptake in 3T3−L1 adipocytes with or without insulin. Rabbit meat extract increased glucose uptake by approximately 2.7 times compared to the control group in insulin−treated 3T3−L1 cells (Figure 6A). In the absence of insulin, the uptake tended to be higher in the rabbit−meat−extract−treated group than in the control group; however, this difference was not statistically significant. To examine the possibility of the Akt pathway being responsible for the rabbit−meat−extract−mediated increase in the glucose uptake, we assessed the activation of Akt phosphorylation. As shown Figure 6B, the rabbit meat extract significantly increased phospho−Akt expression in insulin−stimulated 3T3−L1 adipocytes. However, the Akt pathway is involved in white adipocyte browning. Vitamin D3 inactivates the Akt pathway and affects the browning of white adipocytes [31], and irisin also increases thermogenesis and lipolysis by decreasing Akt phosphorylation in 3T3−L1 adipocytes [32]. In other words, the Akt pathway is known to inhibit the induction of the browning of adipocytes. In this study, no significant change was observed in the expression of Akt phosphorylation in the 3T3−L1 cells not treated with insulin. These results suggestted that rabbit meat extract did not contain factors that can directly activate the Akt pathway. The findings revealed that the rabbit meat extract significantly enhanced insulin−stimulated glucose uptake, improving insulin resistance.

According to the 10th edition of the national standard food composition tables and databases by the Korean Rural Development Administration, the alpha−linolenic acid content in rabbit meat is 20 times higher than that of beef and 2.6 times higher than that of pork. Additionally, the docosahexaenoic acid content is more than 10 times higher in rabbit meat than beef and pork, and the eicosapentaenoic acid content is more than 23 times higher. Alpha−linolenic acid reduces fat mass and fat cell size, shows a high expression of brown fat marker genes, and improves insulin resistance [33,34]. Eicosapentaenoic acid and docosahexaenoic acid have the potential to stimulate the browning of white adipocytes and increase the expression levels of thermogenic markers [35,36,37]. Calcium, phosphorus, and potassium, which prevent obesity, are also present at higher levels than beef and pork [38,39,40,41]. Although specific substances responsible for the conversion of white adipocytes into brown adipocytes in rabbit meat extract are not known, rabbit meat contains higher levels of components that may help to prevent obesity.

## 4. Conclusions

Rabbit meat extract exhibited the induction of fat browning in 3T3−L1 adipocytes in our current study. The browning effect of rabbit meat extract on 3T3−L1 cells decreased the accumulation of lipid droplets and increased brown/beige adipocyte−specific marker genes. This process could depend on the activation of the AMPK pathway. This in vitro study provides experimental evidence to support further research regarding the properties of rabbit meat extract with respect to preventing obesity in in vivo animal studies or clinical trials. Further investigation on the anti−obesity−related bioactive substances derived from rabbit meat extract in the future may provide deeper insights.

## Figures and Tables

**Figure 1 foods-12-03671-f001:**
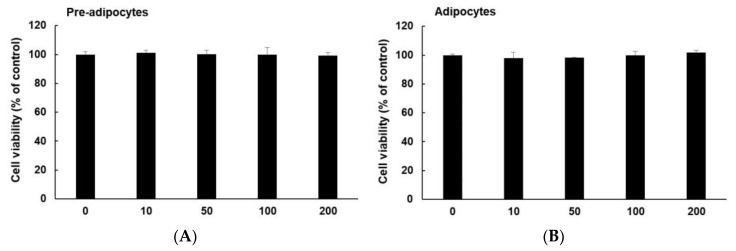
Cell viability of the 3T3−L1 cells in the (**A**) pre−adipocytes and (**B**) adipocytes treated with various concentrations of rabbit meat extract. The data are presented as mean ± SD for three different experiments.

**Figure 2 foods-12-03671-f002:**
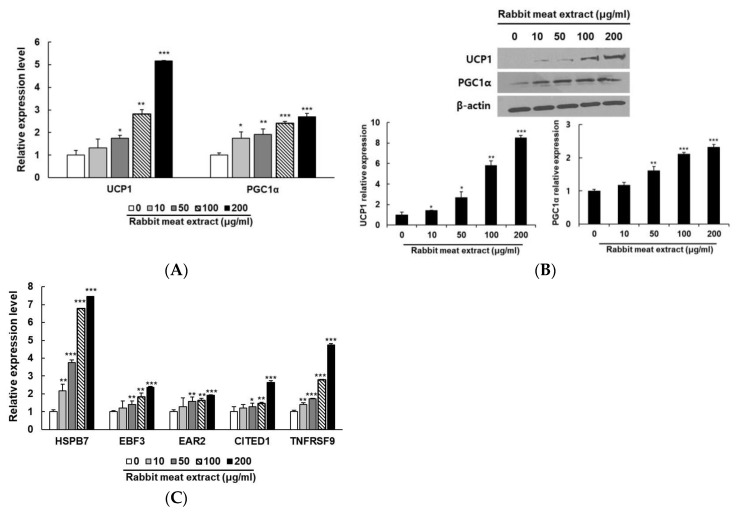
Effect of the rabbit meat extract on the expression of brown−adipocyte−specific markers. Expression of thermogenic genes (UCP1 and PGC1α) was analyzed using (**A**) qRT−PCR analysis (**B**) and Western blot analysis in 3T3−L1 cells treated with rabbit meat extract. * *p* < 0.05, ** *p* < 0.01, *** *p* < 0.001. (**C**) Relative mRNA expression levels of brown−specific genes (HSPB7, EBF3, EAR2, CITED1, and TNFRSF9) were measured using qRT−PCR for different concentrations of rabbit meat extract (10, 20, 50, and 100 μg/mL) in 3T3−L1 adipocytes. * *p* < 0.05, ** *p* < 0.01, *** *p* < 0.001. β−actin mRNA and protein were used as internal controls. The data are presented as mean ± SD for three different experiments.

**Figure 3 foods-12-03671-f003:**
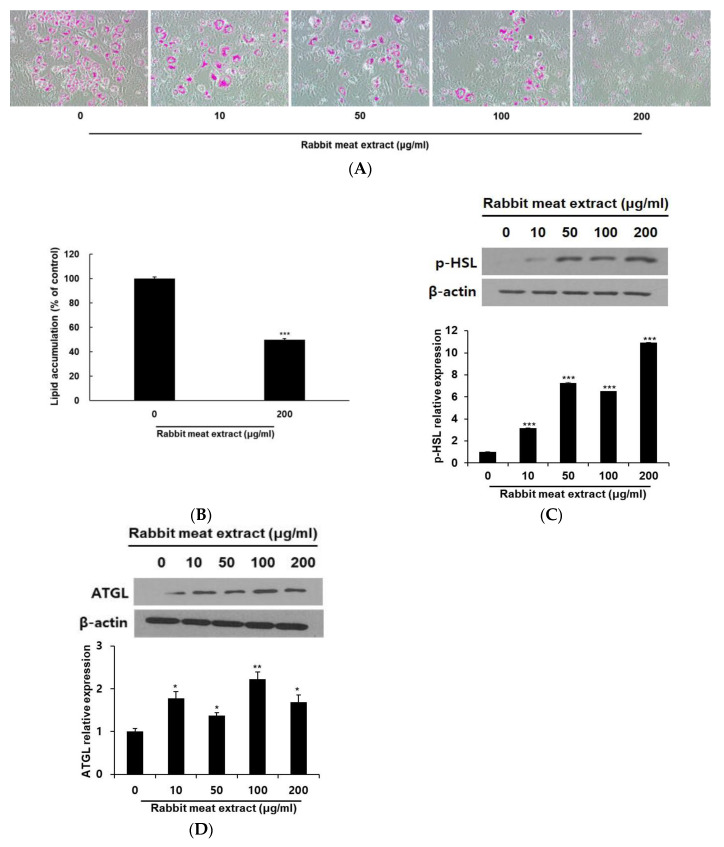
Effect of rabbit meat extract on lipid accumulation and lipolysis−associated protein expressions in differentiated 3T3−L1 cells. (**A**) Cells were subjected to oil red O staining to visualize the lipid droplets using light microscopy (magnification, ×100) and (**B**) quantification by measuring the absorbance at 500 nm. *** *p* < 0.001. (**C**,**D**) The expression of lipolysis−associated genes (hormone−sensitive lipase—HSL and adipose triglyceride lipase—ATGL) in rabbit−meat−extract−treated 3T3−L1 adipocytes was detected using Western blot analysis. * *p* < 0.05, ** *p* < 0.01, *** *p* < 0.001. β−actin was used as a control. The data are presented as mean ± SD for three different experiments.

**Figure 4 foods-12-03671-f004:**
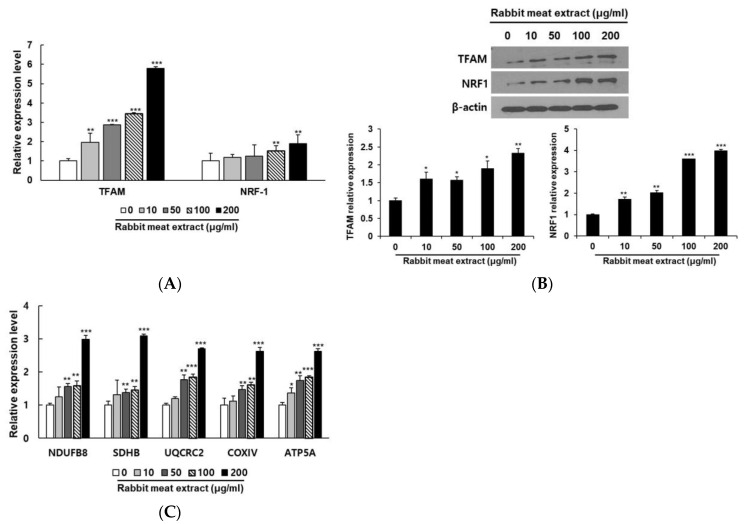
Effect of rabbit meat extract on expression of mitochondrial biogenesis−related proteins. (**A**) mRNA levels of NRF1 and TFAM were determined via qRT−PCR analysis. ** *p* < 0.01, *** *p* < 0.001. (**B**) Protein levels of NRF1 and TFAM were determined using Western blot analysis. * *p* < 0.05, ** *p* < 0.01, *** *p* < 0.001. (**C**) The expression of mitochondrial biogenesis−related genes (NDUFB8, SDHB, UQCRC2, COXIV, and ATP5A) in rabbit−meat−extract−treated 3T3−L1 adipocytes was determined via qRT−PCR analysis. * *p* < 0.05, ** *p* < 0.01, *** *p* < 0.001. β−actin was used as a control. The data are presented as mean ± SD for three different experiments.

**Figure 5 foods-12-03671-f005:**
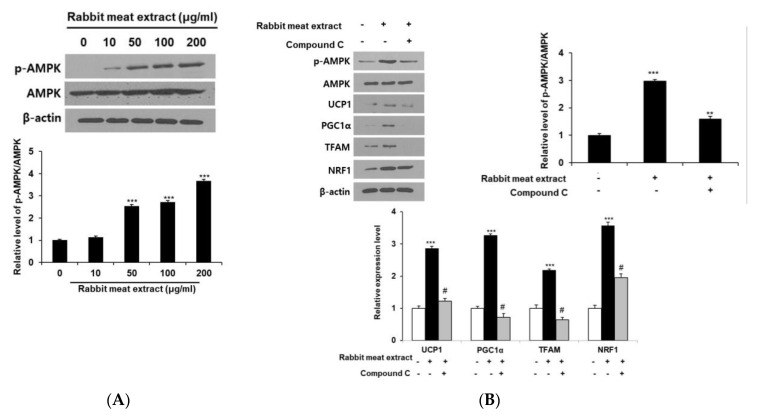
Effect of rabbit meat extract on AMP−activated protein kinase (AMPK) activation in 3T3−L1 adipocytes. (**A**) Western blots were analyzed to determine expression of p−AMPK and AMPK. *** *p* < 0.001. (**B**) Cells exposed to 200 μg/mL rabbit meat extract were treated with the AMPK inhibitor compound C. The level of mitochondrial biogenesis−related genes and thermogenic genes in 3T3−L1 adipocytes were analyzed using Western blot analysis (** *p* < 0.01, *** versus control, *p* < 0.001; white bar, # versus the rabbit−meat−extract−treated cells; black bar, *p* < 0.001). β−actin was used as a control. The data are presented as mean ± SD for three different experiments.

**Figure 6 foods-12-03671-f006:**
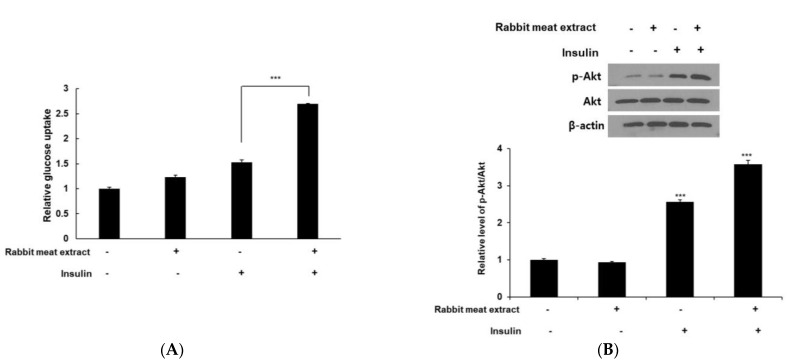
Effect of rabbit meat extract on glucose uptake in 3T3−L1 cells. The glucose uptake (**A**) was determined with 2DG−glucose uptake kit, (**B**) and expression levels of phosphorylated Akt and Akt were detected using Western blotting. The data are presented as mean ± SD for three different experiments. *** *p* < 0.001.

## Data Availability

The data used to support the findings of this study are available from the corresponding authors.

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
