# Peer review of "Rabbit Meat Extract Induces Browning in 3T3−L1 Adipocytes via the AMP−Activated Protein Kinase Pathway"

_foods, 2023, doi:10.3390/foods12193671_

Round 1
Reviewer 1 Report
1.Rabbit species, sampling sites and the main ingredient of rabbit meat extract were not described.
2.The serial number of the figure and the serial number quoted in the text do not have the same case.
3.The discussion about the AMPK pathway is insufficient.
Reviewer 2 Report
Manuscript foods-2623896, entitled “Rabbit Meat Extract Induces Browning in 3T3-L1 Adipocytes via AMPK pathway”
This article provides useful information on the effects of rabbit meat extract on browning in 3T3-L1 Adipocytes via the AMPK pathway. It is in general appropriately organized, carried out and written, however there are some points that should be corrected or clarified.
My main concern is the meat extract that you have used. Composition? Why was this method of extraction selected? Were some of the components of meat lost due to these conditions? And the positive results are an effect of what?
L12-13: “Rabbit meat is of high nutritional value. Although several studies describe the nutritional components of rabbit meat, its role against obesity remains…”
L17: “This study investigated the effects of rabbit meat extract on the 3T3-L1 adipocyte…”
L33: “avoid” instead of “stave off”
L91: “were” instead of “was”
L298: “In a meta-analysis…”
L302: “On the other hand” instead of “Conversely”
L304: “chicken meat” instead of “poultry”
L320-322: Please rephrase
Minor editing of English language required
Reviewer 3 Report
Title: Rabbit Meat Extract Induces Browning in 3T3-L1 Adipocytes via AMPK pathway
The manuscript “Rabbit Meat Extract Induces Browning in 3T3-L1 Adipocytes via AMPK pathway” investigated the functions of rabbit meat extract in the 3T3-L1 adipocyte, focusing on browning induction via analyzing genes specifically expressed in differentiated adipocytes. Study suggest that the rabbit meat extract induces browning of white adipocytes via activation of the AMPK pathway. Therefore, rabbit meat extract is suggested to have potential therapeutic effects for preventing obesity. It is well written article with some interesting findings; however, there are some corrections:
Abstract effectively conveys the main objectives, methods, and key findings of the study. However, it could benefit from a brief mention of the study's limitations and a stronger emphasis on the potential clinical or practical applications of the research. These additions would enhance the abstract's completeness.
Introduction effectively sets the stage for the research by providing a clear rationale for the study, referencing relevant literature, and outlining its specific objectives. It also provides a smooth transition to the research focus, which is the role of rabbit meat extract in inducing browning of adipocytes via the AMPK pathway. However, it could benefit from a brief mention of the study's potential implications or contributions to the field of obesity research.
Line 64: How the rabbit meat was obtained. What was the slaughtering procedure and at what postmortem time meat was taken? As the postmortem metabolic enzymes has different potential at different postmortem time period.
Results:
1. Clearly indicate the statistical significance of your results using appropriate symbols and include p-values where applicable.
2. Provide brief interpretations or explanations alongside each result to help readers understand the implications of your findings.
3. Acknowledge any limitations in your study that may have affected the results and discuss how these limitations were addressed.
4. Include statistical data where relevant, such as means, standard deviations, or confidence intervals, to provide a clearer picture of the results' magnitude and variability.
I would suggest to merge the results and discussion segment in order to make it easy for the readers to understand.
Authors should write the conclusion part with its chances of practical implication and futuristic vision i.e., in future where the scientists should focus.
English grammar and sentence structure should be revised and corrected throughout the manuscript.
English grammar and sentence structure should be revised and corrected throughout the manuscript.
Reviewer 4 Report
In the manuscript foods-2623896 entitled: Rabbit Meat Extract Induces Browning in 3T3-L1 Adipocytes via AMPK pathway”. This study investigated the functions of rabbit meat extract in the 3T3-L1 adipocyte, focusing on browning induction via analyzing genes specifically expressed in differentiated adipocytes. The study is of interest in the field of functional food. The study problem and methodology are generally well justified. The authors should revise their manuscript carefully regarding the following comments.
General comments
- Please describe all abbreviations in their first mention.
- Some graphs must be clear their quality is not good. please rearrange Figures 2 and 4 to be more readable.
- The discussion section is too short, please, discuss your study with other similar studies and please state the superiorities of your study when compared to previous ones.
- There was no conclusion section, please write this section it is extremely important to highlight your recommendations and limitations of your work.
- Please use journal styl for refernceses
Reviewer 5 Report
Rabbit meat is a white meat characterized by low fat and low cholesterol. The authors investigated the anti-adipogenic effect of rabbit meat extract on 3T3-L1 adipocytes and were innovative in understanding the underlying mechanism by focusing on AMPK activity. As a general comment, the authors need to improve some of the detailed presentation to provide a better manuscript. The experimental design of the manuscript needs to be more rigorous, and a conclusion about the results should be added. In addition, the study did not specifically delve into which active ingredients in the rabbit extract had anti-obesity effects. Therefore, I think that this article needs major revisions at this point.
1. Lines 12–16: Please consider rewriting this sentence and conveying only the formative information.
2. Lines 47–49: What active ingredients in Panax ginseng and Diospyros kaki leaf inhibit fat production? Are these active ingredients related to the active ingredients in rabbit meat? Do not make unnecessary references.
3. Line 56: Beef is red meat, while rabbit meat is white meat, is the reference here appropriate?
4. Lines 64–69: Please specify the breed of rabbit and how it was pre-treated.
5. Results and discussion:
a. Figure 1 lacks concentration units for rabbit meat extract.
b. The clarity of Figure 3(a) needs to be increased.
6. Does measuring the phosphorylated AMPK protein levels in 3T3-L1 cells that have been exposed to rabbit meat extract prove that the AMPK pathway is responsible for the effect of the extract on fat cell browning? Do additional indications need to be used to verify the experiment?
7. Please add the conclusion of the article to facilitate the reader's understanding.
The language is generally fluent, with accurate and appropriate use of vocabulary and grammatical structures; the writing is relatively coherent; the writing and punctuation are standardised and neat.
Round 2
Reviewer 3 Report
Thank you for your efforts to improve the manuscript according to the comments and suggestions of the reviewer.
Reviewer 5 Report
The paper can be accepted in this revision.